# SoftStep: Learning Sparse Similarity Powers Deep Neighbor-Based Regression

## Abstract

Neighbor-based methods are a natural alternative to linear prediction for tabular data when relationships between inputs and targets exhibit complexity such as nonlinearity, periodicity, or heteroscedasticity. Yet in deep learning on unstructured data, nonparametric neighbor-based approaches are rarely implemented in lieu of simple linear heads. This is primarily due to the ability of systems equipped with linear regression heads to co-learn internal representations along with the linear head's parameters. To unlock the full potential of neighbor-based methods in neural networks we introduce SoftStep, a parametric module that learns sparse instance-wise similarity measures directly from data. When integrated with existing neighbor-based methods, SoftStep enables regression models that consistently outperform linear heads across diverse architectures, domains, and training scenarios. We focus on regression tasks, where we show theoretically that neighbor-based prediction with a mean squared error objective constitutes a metric learning algorithm that induces well-structured embedding spaces. We then demonstrate analytically and empirically that this representational structure translates into superior performance when combined with the sparse, instance-wise similarity measures introduced by SoftStep. Beyond regression, SoftStep is a general method for learning instance-wise similarity in deep neural networks, with broad applicability to attention mechanisms, metric learning, representational alignment, and related paradigms.

## 1 Introduction

Classical machine learning offers a wide variety of algorithms for supervised regression on tabular data. Methods such as neighbor-based regression, decision trees, and kernel support vector machines are well-known to outperform linear regression when the underlying regression manifold is nonlinear Araghinejad (2013). In contrast, contemporary deep learning pipelines almost universally employ a linear layer as the terminal predictor. While this linear assumption is restrictive, its deficiencies are often mitigated by the capacity to co-learn both internal representations and output regression parameters with gradient descent, which allows deep networks to adapt their extracted features to a simple linear predictor Monasson and Zecchina (1995).

Alternative regression paradigms, however, remain underutilized in deep learning. Many traditional methods are non-differentiable, preventing seamless integration with gradient-based training, while others are nonparametric, imposing rigid inductive biases on representational geometry that may conflict with effective feature learning. Empirical evidence from applied domains such as biomedical prediction suggests that if classical regression algorithms could be embedded into end-to-end differentiable systems with learnable parameters, they could surpass linear heads by simultaneously relaxing distributional assumptions and retaining the benefits of joint representation–predictor optimization Pan et al. (2019); Zolnoori et al. (2023).

To address this gap, we introduce SoftStep, a drop-in module for neural networks that learns sparse, instance-wise similarity measures within neural embedding spaces. When combined with neighbor-based methods such as $k$-nearest neighbors ($k$-NN) or neighborhood component analysis (NCA) (Cover and Hart, 1967; Goldberger et al., 2004), SoftStep enables differentiable regression heads that bring the expressive power of nonlinear predictors into the deep learning setting. Our theoretical analysis shows that neighbor-based regression with a mean squared error (MSE) loss implicitly

Figure 1: **The architecture of the neural network regressors tested in this paper.** The left side shows the head-to-head comparison between linear heads and neighbor-based heads evaluated in this work. The right side shows the specifics of neighbor-based regression augmented with SoftStep. Here, $\sim$ refers to a soft ranking operation in the context of differentiable $k$-NN and a row-wise min-max normalization of the similarity matrix in the context of NCA.

enforces pairwise and triplet constraints on the embedding geometry, promoting well-structured and predictive representations. Moreover, SoftStep strengthens these geometries by adaptively weighting similarities, yielding superior predictive performance compared to standard linear regression heads. Crucially, our goal is not to claim SoftStep as a state-of-the-art method in all contexts, but rather to demonstrate that **neighbor-based regression augmented with SoftStep can serve as a plug-and-play alternative to linear regression in diverse architectures.**

Across experiments, we pair large-scale feature extractors (e.g., convolutional neural networks, transformers) with intermediate decoding multilayer perceptrons (MLPs), and consistently observe performance gains when substituting linear output heads with SoftStep neighbor-based regression heads. Our contributions are as follows;

- A novel module: We introduce SoftStep, a family of step-like functions which learns instance-wise similarity measures for neighbor-based regression in deep neural networks.
- Theoretical insight: We provide an optimization analysis showing that neighbor-based regression with MSE induces implicit pairwise and triplet regularization on learned embeddings.
- Empirical validation: We demonstrate that SoftStep-equipped regression heads outperform linear predictors across multiple unstructured data domains.

A GitHub repository containing the full implementation of our methods and all experimentation is available to the community at the SoftStep GitHub repository.

## 2 RELATED WORK

This work is in part motivated by the success of non-linear prediction - particularly neighbor-based models - in both classical and modern machine learning. Our methods resemble techniques used in representation learning, sparse attention, and graph attention. Our analyses show the viability of our prediction algorithm to act as a supervised metric learning algorithm similar to contrastive and triplet-loss methods. As such, we include an overview of relevant literature in these areas.

**Neighbor-based models**   Neighbor-based models make predictions by associating data points according to a similarity or distance measure Fix (1985); Cover and Hart (1967). The canonical example is $k$-nearest neighbors ($k$-NN), along with its modern extensions. Differential Nearest Neighbors Regression (DNNR) leverages neighbors-of-neighbors to approximate local gradients of the regression manifold (Nader et al., 2022), but requires solving a separate least-squares problem for each neighbor of the query point, making it incompatible with efficient forward passes in deep networks. Neighborhood Component Analysis (NCA) offers a more natural alternative (Goldberger et al., 2004), weighting neighbor labels according to similarity scores rather than ranks. Deep learning systems that explicitly incorporate pairwise similarities have also emerged, including SAINT for tabular data via row-wise attention (Somepalli et al., 2021), neural methods for image restoration (Plötz and Roth, 2018), confidence estimation (Papernot and McDaniel, 2018), and collaborative

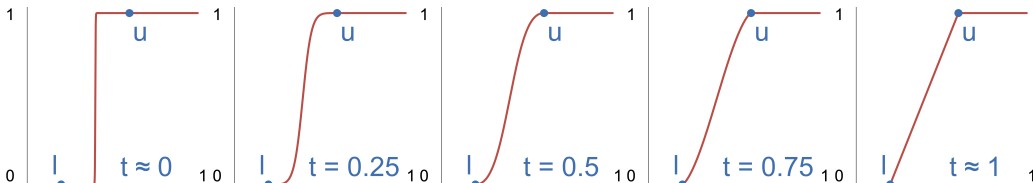

Figure 2: **The SoftStep family of functions learns sparse attention smoothly and differentiably.** SoftStep is a family of increasing surjective functions mapping the unit interval to itself. Plotted here is the SoftStep function as the transition parameter $t$ varies from 0 to 1. Parameters $l$ and $u$ define the transition boundaries.

filtering He et al. (2017). In few-shot learning, matching networks exploit sample associations through pretrained embeddings Vinyals et al. (2016). Despite these successes, general frameworks for deep neighbor-based regression remain underexplored, with most existing approaches tied to specific tasks or domains.

**Representation Learning** Metric learning seeks to construct embedding spaces where semantic relationships between data points are captured by distances or similarity measures Yang and Jin (2006). Classical approaches such as NCA and Large Margin Nearest Neighbor (LMNN) (Weinberger et al., 2005) learn transformations that enforce class-consistent neighborhoods, while extensions such as Metric Learning for Kernel Regression (MLKR) adapt these ideas to regression Weinberger and Tesauro (2007). Embedding similarity also underlies several adjacent areas. In representational alignment, pairwise and higher-order structures guide comparisons across models or between artificial and biological systems Kriegeskorte et al. (2008). More recent approaches in this area have pointed towards the promise of learning sparse similarity measures for alignment Anonymous. In graph learning, graph attention networks induce edge weights between nodes, enabling relational reasoning Veličković et al. (2018).

**Sparse Attention** In Section 3 we show how SoftStep can be thought of as a mechanism for learning a sparse adaptive attention mechanism, that we employ in the context of prediction. Sparse attention is concerned with reducing the number of inputs attending to each other within a neural attention mechanism Tay et al. (2020). Besides reducing compute complexity, the primary motivation behind these approaches is to develop mechanisms that are more selective than standard attention. Traditional attention relies on the SoftMax function, which yields non-zero attention weights for all inputs Vaswani et al. (2017a). A number of methods have been introduced to account for this, such as top-k attention (Gupta et al., 2021), local windowing (Nguyen et al., 2020), Sparsek (Lou et al., 2024), and Reformer-based methods Kitaev et al. (2020). While these approaches limit the number of nonzero attention weights, many are non-differentiable. Consequently, they cannot be optimized effectively in the context of end-to-end deep learning. More sophisticated methods, such as Sparsemax (Martins and Astudillo, 2016), produce sparse outputs via a non-learned projection, but do not adapt to the semantic context of local densities of the embedding space.

**Nonlinear Prediction on Neural Network Embeddings** In some applications, particularly in computational biomedicine, pretrained neural networks are treated as feature extractors for traditional machine learning algorithms such as $k$-NN and support vector machines Notley and Magdon-Ismail (2018). In small- to mid-size data regimes, where training a neural network from scratch will lead to poor generalization, supervised finetuning of a pretrained neural network can improve performance on data domains of interest Tajbakhsh et al. (2016); Spolaôr et al. (2024). This two-stage process - supervised finetuning of a pretrained neural network followed by fitting extracted features from the finetuned model on different classification and regression algorithms - has seen widespread success in various tasks, such as prediction of bone age from radiology images or predicting Alzheimer's disease from speech recordings Pan et al. (2019); Luz et al. (2021a); Balagopalan and Novikova (2021); Laska et al. (2024); Zolnoori et al. (2023); Nachesa and Niculae (2025). This body of literature suggests that despite internal representations of data within a neural network being optimized to fit a linear output head, further performance gains can be unlocked using nonlinear prediction heads.

## 3 METHODS

### 3.1 SOFTSTEP - A PARAMETRIC FUNCTION FOR SIMILARITY WEIGHTING

Given a deep learning regression architecture, we denote by $F$ the feature extractor, i.e., all components upstream of the regression head. For data–label pairs $(X, y) \subset \Omega \times \mathbb{R}$, where $\Omega$ is the sample space, the feature extractor maps the inputs into an embedding space via $F(X) = Z \subset \mathbb{R}^d$ with embedding dimension $d$. For an embedded sample $z^* = F(x^*)$, we define $SoftStep_{z^*}$ as a transformation that reshapes the similarities to $z^*$ according to

$$SoftStep_{z^*}(s) = \begin{cases} 0, & 0 \leq s \leq l, \\ \dfrac{(s-l)^{1/t}}{(s-l)^{1/t} + (u-s)^{1/t}}, & l < s < u, \quad \text{where } (l, u, t) = \sigma\big(W_{ss}(z^*) + b_{ss}\big). \\ 1, & u \leq s \leq 1, \end{cases}$$

(1)

The parameters $(l, u, t)$ are learned as a functions of $z^*$ and determine a mapping of similarities to the unit interval $[0, 1]$. Although $SoftStep$ is piecewise-defined, it is differentiable on its domain, including at the boundary points $l$ and $u$. The parameter $l$ sets the lower similarity threshold below which values are mapped to 0, thereby controlling the sparsity of $SoftStep_{z^*}$. Conversely, $u$ defines the upper threshold above which similarities are mapped to 1, while $t$ governs curvature of the transition between these two extreme regimes. Taken together, the parameters yield a sigmoid-like curve that flexibly adapts to each embedded sample, achieving exact zeros and ones over learned regions of the similarity domain.

Notably, SoftStep can also be implemented with model-level, i.e., global, parameters. In this case similarities to every embedded point are warped with respect to the same learned parameter combination. This formulation of the module may be useful in the small-data regime where the potential for overfitting precludes the usefulness of the instance-wise formulation. We note that the SoftStep function family is similar to SmoothStep functions which have been widely adopted in computer graphics Hazimeh and Mazumder (2020). SoftStep is distinct due to curvature on the transition domain being governed by a continuous parameter which can be learned through gradient descent. A visualization of SoftStep under different parameterizations is provided in Figure 2.

### 3.2 DIFFERENTIABLE $k$-NEAREST NEIGHBORS

A natural regression model to pair with SoftStep is $k$-nearest neighbors ($k$-NN). We construct a differentiable variant using the `torchsort` package (Blondel et al., 2020), which provides approximate differentiable rankings of neighbor similarities. For query points $Z = F(X)$ and neighbor set $X_N \times y_N = \{(x_n, y_n)\}_{n=1}^N$, we define a similarity matrix $Sim(Z, Z_N)$ under a chosen similarity measure like negative $\ell_2$ distance, cosine similarity, or radial basis function (RBF) kernel. Differentiable ranks are then computed as

$$Rank(Z, Z_N) = \texttt{soft\_rank}(Sim(Z, Z_N)),$$

and similarities are transformed according to

$$Sim(Z, Z_N) \quad \leftarrow \quad Sim(Z, Z_N) + \ln\big(SoftStep(Rank(Z, Z_N))\big).$$

Predictions are obtained as

$$\hat{y} = SoftMax(Sim(Z, Z_N)) \, y_N.$$

In this formulation, the $SoftStep$ parameter $u$ acts as a learned, instance-specific $k$ that controls the effective neighborhood size, while the other parameters are held fixed at stable values. In $k$-NN terminology, this algorithm corresponds to a data-adaptive weighting scheme over the $k$ nearest neighbors. Further implementation details, such as masking self-similarities and configuring `soft_rank`, are provided in Algorithm 1.

### 3.3 NEIGHBORHOOD COMPONENT ANALYSIS

Neighborhood component analysis (NCA) provides an alternative neighbor-based regression algorithm by operating directly on raw similarities rather than ranks of similarities. To integrate SoftStep

---

**Algorithm 1** Differentiable $k$-NN with SoftStep

---

1: **procedure** DIFFKNN-SOFTSTEP($Z, Z_N, param\_mode$)
   **Initialization (run once at module construction):**
2:    **if** $param\_mode$ = "instance_wise" **then**
3:        $k\_param \leftarrow$ Linear layer with sigmoid activation
4:    **else if** $param\_mode$ = "global" **then**
5:        $k\_param \leftarrow$ Learnable scalar model-level parameter
6:    **end if**
   **Forward Pass:**
7:    $sim \leftarrow$ SIM($Z, Z_N, similarity$)                                  ▷ Matrix of similarities
8:    **if training then**
9:        $N \leftarrow |Z_N|$                                                  ▷ Neighbor set size
10:       $sim \leftarrow sim -$ DIAG($\infty$)                              ▷ Push self-similarity to worst rank
11:   **end if**
12:   $ranks \leftarrow$ SOFT_RANK($-sim$) $- 1.0$                     ▷ Rank neighbors per query
13:   **if** $param\_mode$ = "instance_wise" **then**
14:       $k \leftarrow \sigma(k\_param(Z)) \cdot (N - 1)$
15:   **else if** $param\_mode$ = "global" **then**
16:       $k \leftarrow \sigma(k\_param) \cdot (N - 1)$
17:   **end if**
18:   $k \leftarrow \max(k, 1)$                                         ▷ Clamp to at least one neighbor per-query
19:   $g \leftarrow$ SOFTKSTEP($ranks, k, q = 4, t = 0.75$)        ▷ Adjusted SoftStep (defined below)
20:   **return** $sim + g$
21: **end procedure**

22: **function** SOFTKSTEP($ranks, k, q, t$) ▷ Flipped SoftStep function for scaling similarities based
    on rankings. $q$ is a fixed hyperparameter controlling the minimum number of neighbors.
23:   $num \leftarrow (k + q - ranks)^{1/t}$
24:   $den \leftarrow num + (k - ranks)^{1/t}$
25:   $step \leftarrow num/den$
26:   $\forall\, i:$ **if** $ranks_i < k$ **then** $step_i \leftarrow 1$
27:   $\forall\, i:$ **if** $ranks_i > k + q$ **then** $step_i \leftarrow 0$
28:   **return** $\log(step)$
29: **end function**

---

into NCA, we apply the transformation

$$Sim(Z, Z_N) \quad \leftarrow \quad Sim(Z, Z_N) + \ln\big(SoftStep(\overline{Sim(Z, Z_N)})\big),$$

where $\overline{Sim(Z, Z_N)}$ denotes the row-wise min–max normalized similarity matrix. Predictions then follow the same SoftMax weighting scheme as in the differentiable $k$-NN variant. Full implementation details are given in Algorithm 2.

**Connection to attention.**   NCA is mechanistically similar to the attention mechanism: both compute a similarity matrix between two sets of embeddings (in attention, queries and keys), normalize each row with a SoftMax operation to obtain weights, and apply those weights to a set of values (in NCA, the neighbor labels). The success of SoftStep in learning sparse similarity measures within NCA suggests that it may also serve as a general mechanism for sparse attention.

**SoftStep in neighbor-based regression**   In both differentiable $k$-NN and NCA, SoftStep acts by shifting similarities by values in the interval $[0, -\infty)$ prior to SoftMax, which in turn rescales them onto $[0, 1]$ while transforming them into usable weights for prediction. This mechanism allows irrelevant neighbors for each instance to be assigned vanishing weights while preserving proportional weighting among label-consistent neighbors. The transition parameter $t$ controls the sharpness of the boundary between excluded and included neighbors, allowing the model to interpolate smoothly between hard selection and soft weighting of neighbor similarities.

---

**Algorithm 2** NCA with SoftStep

---

1: **procedure** SOFTSTEP($Z, Z_N, param\_mode$)
    **Initialization (run once at module construction):**
2:    **if** $param\_mode$ = "instance_wise" **then**
3:        $params \leftarrow$ Linear layer with sigmoid activation
4:    **else if** $param\_mode$ = "global" **then**
5:        $params \leftarrow$ Learnable model-level parameters
6:    **end if**
    **Forward Pass:**
7:    **if** $param\_mode$ = "instance_wise" **then**
8:        $(l_0, u_0, t) \leftarrow params(Z)$
9:    **else if** $param\_mode$ = "global" **then**
10:       $(l_0, u_0, t) \leftarrow \sigma(params)$
11:    **end if**
12:   $sim \leftarrow$ SIM($Z, Z_N$)                      ▷ Matrix of similarities
13:   $sim_{\text{norm}} \leftarrow$ NORM($sim$)               ▷ Normalize to $[0, 1]$
14:   $top\_sim \leftarrow$ ROWMAXEXCLSELFSIM($sim_{\text{norm}}$)
15:   $l \leftarrow \min(l_0, top\_sim) - \epsilon$               ▷ $\epsilon > 0$ small
16:   $u \leftarrow l + u_0 \cdot (1 - l)$
17:   **return** $sim + \log($SOFTSTEP($sim_{\text{norm}}, l, u, t$)
18: **end procedure**

---

## 4 REPRESENTATIONAL GEOMETRY OF NEIGHBOR-BASED REGRESSION

### 4.1 NEIGHBOR-BASED REGRESSION WITH MSE OBJECTIVE STRUCTURES REPRESENTATIONS

Here, we demonstrate that a neighbor-based regression model paired with MSE loss yields implicit optimization conditions for structuring pairs of points in the embedding space with respect to their labels, as well as conditions for structuring triplets of points. We define $p_{ij} = \frac{e^{s_{ij}}}{\sum_{k=1}^{N} e^{s_{ik}}}$ where $s_{ij} = Sim(Z, Z_N)_{ij}$ with similarity warping due to SoftStep. Furthermore, $\Delta_{ij} = y_i - y_j$. We observe that,

$$\mathcal{L}(y, \hat{y}) = \frac{1}{N} \sum_{i=1}^{N} (y_i - \sum_{j=1}^{N} p_{ij} y_j)^2 = \frac{1}{N} \sum_{i=1}^{N} \left( \sum_{j=1}^{N} \Delta_{ij} p_{ij} \right)^2$$

due to $\sum_{j=1}^{N} p_{ij} = 1$. Terms of the form $\Delta_{ij}^2 p_{ij}^2$ represent a pairwise similarity loss on the points indexed by $(i, j)$. Assigning $j$ high proportional similarity to $i$ accrues a penalty as an increasing function of the label distance - **MSE loss is explicitly minimized when dissimilarly labeled points repel one another.**

Triplet terms $\Delta_{ij} p_{ij} \Delta_{ik} p_{ik}$ also encode embedding conditions. However, a more illuminating description of the optimal triplet embedding geometry considers the partial optimization of terms of the form:

$$T_{ijk} = (\Delta_{ij} p_{ij} + \Delta_{ik} p_{ik})^2 \quad \text{subject to} \quad p_{ij} + p_{ik} = 1 - \sum_{l \neq j, k} p_{il} = R$$

This optimization analysis will yield different embedding conditions depending on the order of the labels $j$ and $k$ relative to the anchor point $i$. In the analysis of this local subproblem, we assume that $p_{il}$ is fixed for all $l \neq j, k$ and that $p_{ij}, p_{ik}$ are optimized independently. We also limit our focus to the case where $y_i \neq y_j \neq y_k$ as the triplet loss collapses into a simpler embedding condition if any of the labels are equal.

We proceed by minimizing $T_{ijk}$ as a convex quadratic with respect to $p_{ij}$ on the interval $[0, R]$.

$$T_{ijk}(p_{ij}) = (\Delta_{ij} p_{ij} + \Delta_{ik}(R - p_{ij}))^2 = ((\Delta_{ij} - \Delta_{ik}) p_{ij} + \Delta_{ik} R)^2$$

has the unconstrained minimizer $p_{ij}^* = \frac{\Delta_{ik}}{\Delta_{ik} - \Delta_{ij}} R$. From here we examine two cases based on the ordering of the three labels $y_i, y_j, y_k$ that will yield two conditions depending on whether the anchor point label lies between the other two labels or is extreme itself.

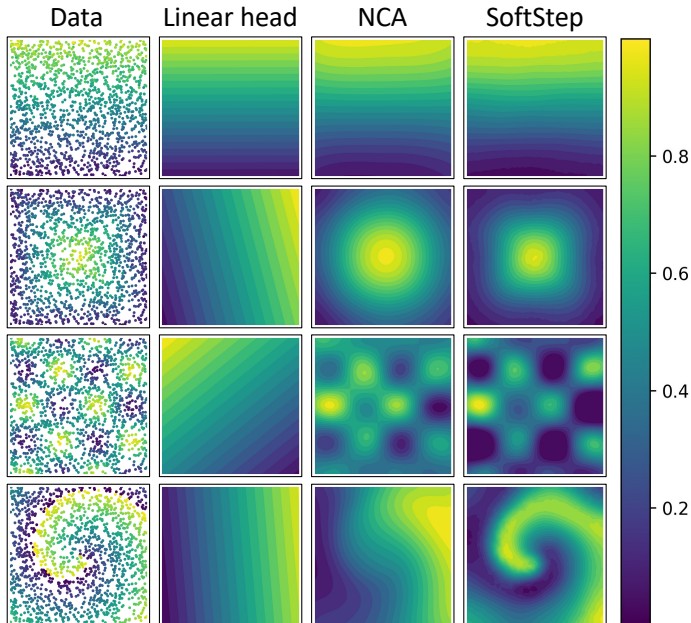

Figure 3: **NCA and SoftStep enhance regression by modeling nonlinear relationships, allowing learned manifolds to better reflect underlying data distributions.** Synthetic datasets were generated by sampling points from the domain $[-1, 1]^2$ and assigning continuous labels using various target functions. The color gradients depict the target values and resulting regression surfaces learned by a linear regression head, NCA 3.3, and NCA augmented with SoftStep 3.1. The linear head cannot capture the complexity of the non-linear targets. Conversely, NCA, especially when augmented with SoftStep, learns smooth nonlinear regression surfaces that capture the underlying distribution. This capability relaxes constraints on upstream neural networks when learning predictive representations.

**Extreme anchor label:** If $y_i < y_j < y_k$, then $p_{ij}^* > R$ and so $T_{ijk}$ is minimized when $p_{ij} = R$, which implies $p_{ik} = 0$. That is, if the label of $j$ closer to that of $i$ than the label of $k$ is, then all the available proportional similarity between their embeddings is optimized to go to $j$. It can be seen that all arrangements of $y_i, y_j, y_k$ in which $y_i$ is extreme yield analogous conditions.

**Intermediate anchor label:** If $y_j < y_i < y_k$ then $0 < p_{ij}^* < R$ and thus is the true minimizer. The condition $p_{ik} = R - p_{ij}$ yields that $T_{ijk}$ is minimized when $p_{ik} = \frac{\Delta_{ij}}{\Delta_{ij} - \Delta_{ik}} R$ and so optimality is achieved when

$$\frac{p_{ij}}{p_{ik}} = \frac{y_k - y_i}{y_i - y_j},$$

In other words, **optimality is achieved when embedding similarity is distributed in *exact proportion* to label similarity.** The same expression results from the case where $y_k < y_i < y_j$. Notice that the above analysis on triplets can be generalized to a quadratic program in as many as $N$ dimensions, which suggests that higher order structures in the embedding space may be jointly optimized for as well. However, as we demonstrate in Appendix C, the solution to this more general optimization problem reduces to the triplet case: the available similarity is optimally allocated to the one or two neighbors whose labels are closest to the anchor's, in proportion to their label differences.

This analysis demonstrates how neighbor-based methods are equipped to learn well-structured and predictive embedding spaces. It is clear that models which admit sparse similarity measures can better optimize for the uncovered structuring conditions in a dense feature space. Furthermore, the ability to learn asymmetric instance-wise similarity measures allows the model to account for differing local densities, enabling improved performance on outlier embeddings and samples with extreme labels, as well as accounting for varying noise-levels across feature space.

| Dataset | Linear | NCA | NCA g | NCA i | DiffKNN g | DiffKNN i | Sparsemax |
|---|---|---|---|---|---|---|---|
| RSNA | 5.12±0.608 | 4.05±0.356 | 4.02±0.317 | **3.78±0.36** | 3.94±0.176 | 4.02±0.415 | 4.01± 0.166 |
| MedSegBench | 6.69±1.29 | 4.87±1.49 | 3.72±0.786 | 10.8±6.41 | 3.82±0.405 | **3.04±0.428** | 3.97±0.626 |
| ADReSSo | 96.2±33.3 | 31.7±6.01 | **24.2±3.50** | 32.3±6.01 | 27.3±3.64 | 25.5±3.42 | 28.1±3.74 |
| CoughVid | 39.7±27.0 | **21.3±0.201** | 21.3±0.217 | 21.3±0.232 | 22.5±0.375 | 22.7±0.401 | 21.5±0.26 |
| NoseMic | 7.96±0.947 | 6.29±1.27 | 5.93±0.735 | 6.19±0.460 | 5.49±0.624 | **5.40±0.533** | 5.59±0.649 |
| Udacity | 0.435±0.170 | 0.041±0.006 | 0.080±0.019 | 0.088±0.011 | 0.040±0.014 | 0.034±0.009 | **0.031±0.0049** |
| Pitchfork | 69.6±167 | 11.2±1.78 | 11.5±2.18 | 12.4±1.79 | **11.1±2.09** | 12.2±1.40 | 11.8±1.65 |
| Houses | 303±81.3 | 14.8±2.60 | **13.4±2.07** | 18.4±2.11 | 15.2±2.14 | 13.7±2.01 | 14.7±2.80 |
| Books | 8.33±0.680 | 7.63±0.575 | 7.64±0.568 | 7.83±0.564 | 7.39±0.476 | **7.12±0.565** | 7.61±0.468 |
| Austin | 2.30±0.216 | 2.09±0.184 | 2.08±0.166 | 2.08±0.197 | **2.01±0.173** | 2.03±0.191 | 2.04±0.18 |
| Wiki-IMDB[†] | 2.51±0.843 | 2.29±0.844 | 5.74±6.72 | 2.68±1.01 | 5.48±8.43 | **2.04±0.593** | 23.6±1.24 |
| Wiki-IMDB | 0.418±0.446 | 0.348±0.764 | 0.448±0.23 | 0.409±0.112 | **0.246±0.061** | 0.256±0.053 | 0.364±0.036 |

Table 1: **SoftStep improves the performance of neighbor-based regression and outperforms the linear regression head across all datasets. The adaptive learned sparsity of SoftStep yields improved sparse neighbor-based regression over the sparsemax benchmark.** Average mean squared error (MSE)±standard deviation across methods over ten random splits of each dataset. The best mean results for each dataset are in bold. Here,"i" denotes a neighbor-based method with instance-wise SoftStep parameters and "g" denotes a method with globally learned SoftStep parameters. While $k$-NN cannot be implemented faithfully as a regression head without SoftStep, NCA can and so is included as well. Wiki-IMDB[†] refers to training the Wiki-IMDB backbone from scratch. All measurements have been scaled by $10^3$ for readability. A complete description of each dataset, along with all preprocessing pipelines and feature extractors can be found in Appendix B.

## 4.2 Embedding Space Constraints of Linear Regression Heads

Linear prediction assumes the existence of a data embedding in which representations are related to the target variable linearly with an error-minimizing constant variance Montgomery et al. (2021). This strong constraining assumption limits the expressiveness of the embedding space that the feature extractor can construct. As a result, rich structures such as multiplicative feature interactions (Batista et al., 2024), threshold effects (Su et al., 2017), and curved manifolds like sinusoidal or spiral relationships cannot be captured by a simple affine map Liang et al. (2020). However, differentiable neighbor-based regression algorithms free the upstream feature extractor to embed data in more complex arrangements that may better preserve its predictive signal. In Figure 3 we demonstrate empirically that NCA is capable of capturing nonlinear relationships between data and a target variable, and that SoftStep sparsity combined with NCA can help adapt the regression manifold to complex local label distributions.

## 5 Experiments

We measure the regression performance of differentiable $k$-NN and NCA, as enabled by SoftStep, using several unstructured datasets for regression. To attain a complete analysis of our methods, we tune the instance-wise parameters in SoftStep and substitute in model-level, or "global" parameters, as detailed in Algorithms 1 and 2. To evaluate performance, each unstructured dataset is paired with a pretrained neural network feature extractor and finetuned in a supervised manner. All datasets are split 80/20 into development and testing, with the development set further split 85/15 into training and validation. Early stopping is performed when MSE performance does not improve on the validation set for 10 epochs. This procedure is repeated 10 times to get an average performance of each model. All training was conducted on a high performance compute cluster equipped with NVIDIA H100 Tensor Core GPUs, with each model trained individually on each fold of data using a single NVIDIA H100 SXM5 80GB GPU for a maximum of 48 hours Kovatch et al. (2020).

Downstream of each feature extractor, a multilayer perceptron with hidden dimension 200 projects samples to an embedding space with dimensionality determined via hyperparameter tuning A, followed by one of the following regression heads: a linear layer; NCA; NCA with instance-wise SoftStep; NCA with global SoftStep; differentiable $k$-NN with instance-wise SoftStep; differentiable $k$-NN with global SoftStep; NCA with sparsemax (Martins and Astudillo, 2016) in place of softmax for sparse similarity/attention scores. Method-specific batch sizes and similarity measures were also tuned for.

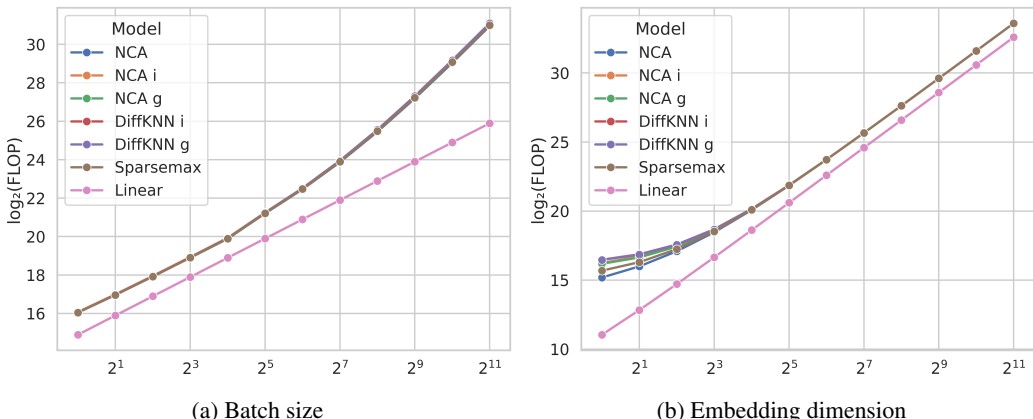

(a) Batch size                  (b) Embedding dimension

Figure 4: We plot the total floating point operations (FLOPs) for each method tested as we increase the batch size/number of neighbors ($N$) and embedding dimension ($d$) exponentially. For each method, we attach a linear layer that condenses the data from $4 \times d$ to $d$ before regression. We run 10 forwards and backwards passes with NVIDIA Tesla V100 PCIE 16GB GPUs and report the average FLOPs. When we vary $N$, $d = 25$ to reflect our experiments. Similarly, when we vary the embedding dimension, $N = 32$. We plot $\log_2(\text{FLOP})$ to account for the exponential scale of the x-axes. Since SoftStep parameter generation and similarity warping are each $O(dN)$, our neighbor-based methods have complexity $O(dN^2)$ as with other self-attention-style mechanisms Vaswani et al. (2017b). Our observations are consistent with Blondel et al. (2020).

In order to demonstrate the viability of our methods on large datasets, we include results on the 100,000+ sample Wiki-IMDB dataset B. For this dataset specifically, we train the backbone model from scratch in addition to the usual supervised finetuning to demonstrate the viability of our methods in the large data regime.

## 6 RESULTS AND DISCUSSION

Our results demonstrate that neighbor-based regression methods, when augmented with the SoftStep module, achieve superior predictive performance compared to both their unaugmented counterparts and traditional linear regression heads. These findings support our theoretical analysis, which showed that neighbor-based regression with mean squared error implicitly imposes geometric constraints on embeddings that become more effectively realized when similarity measures are sparse. SoftStep provides precisely this functionality by adaptively reshaping similarities into sharp inclusion–exclusion regimes that can adapt to local density and label structure.

A key contribution of this work is demonstrating that neighbor-based regression heads need not be relegated to shallow models or post-hoc feature extraction pipelines, but rather can be integrated directly into end-to-end deep learning systems as trainable and competitive alternatives to linear prediction heads. Our results indicate that this integration not only avoids the performance degradation that traditionally accompanies nonparametric methods in neural networks, but can in fact surpass the performance of linear heads across diverse architectures and domains. SoftStep also offers an important conceptual bridge between neighbor-based regression and other areas of machine learning. Mechanistically, SoftStep bears close similarity to sparse attention: both compute similarity-based weights over a set of candidates, but SoftStep introduces explicit sparsity and adaptivity that allow irrelevant neighbors to receive zero weight. This connection suggests that SoftStep may provide a general-purpose mechanism for selective attention in settings where full SoftMax weighting is undesirable. Furthermore, the ability of SoftStep to learn instance-wise similarity transformations resonates with recent advances in representational alignment Anonymous. Our framework thus contributes to a growing set of tools for learning and interpreting representational geometry in high-dimensional spaces.

Looking ahead, we see promising opportunities for combining linear and neighbor-based prediction paradigms in heterogeneous ensembles. Linear heads offer unmatched efficiency and global stability,

while SoftStep-augmented neighbor heads provide local adaptivity and nonlinear expressiveness. By co-training these predictors within a single model, it may be possible to capture both global and local patterns in the data, yielding systems that are simultaneously robust, expressive, and efficient. Future work will extend our analyses to such ensembles, as well as to broader task domains and architectural settings.

Despite these advantages, there are important limitations to acknowledge. First, while our experiments span multiple domains, the evaluation remains restricted to regression tasks and a limited set of architectures. Second, the introduction of neighbor-based heads carries computational overhead relative to linear layers (Figure 4), particularly in settings with large batch sizes or high-dimensional embeddings. Although differentiable ranking and similarity warping are efficient in practice, scaling to extremely large datasets will require further optimization. Finally, while we provide theoretical and empirical evidence of improved embedding geometry, the precise dynamics of how SoftStep interacts with upstream feature extractors requires further exploration. Due to our focus on neighbor-based regression, it remains to be established whether SoftStep's benefits extend as strongly to other relevant areas of deep learning.

## 7    CONCLUSION

This work introduces SoftStep, a parametric module that learns sparse, instance-wise similarity measures within neural networks. By augmenting neighbor-based regression heads such as differentiable $k$-NN and neighborhood component analysis, SoftStep enables predictive performance that consistently exceeds that of linear regression heads. An exploration of the representational geometries induced by these methods underscores the utility of SoftStep and the potential for future research exploring these geometries and realizing their functional potential. Beyond empirical improvements, SoftStep highlights deep connections between neighbor-based methods, sparse attention, and representational alignment, offering a unifying perspective on how local similarity structure can be exploited in neural embedding spaces.

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

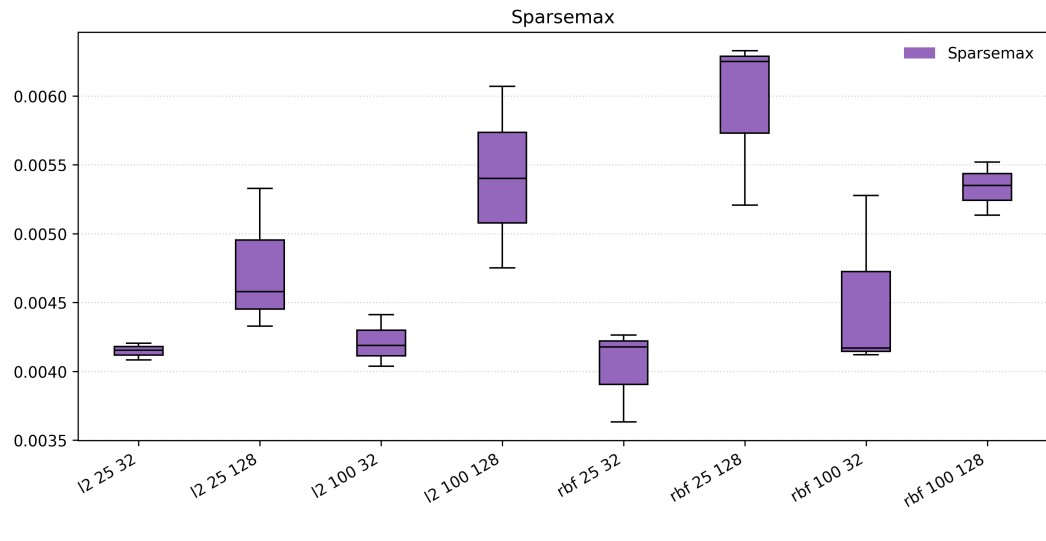

Figure 5: The x-axis ticks are hyperparameter combination triplets of the form (similarity measure, embedding dimension, batch size). The best-performing hyperparameter configurations by mean performance were (RBF, 25, 32).

## A APPENDIX: HYPERPARAMETER TUNING

We conduct hyperparameter tuning on both the neighbor-based and dense layer prediction-based models used in our evaluation. We use the RSNA (Halabi et al., 2019) dataset to determine hyperparameters and then apply the resulting configurations for experiments on all other datasets. In our tuning we evaluate on 3 random splits of the data for every configuration, whereas in the full experimentation we evaluate on 10. All MSE scores below are scaled by $10^3$ for readability.

We conduct a grid search over the space of batch sizes, and embedding dimensions. For our neighbor-based models, we also search over similarity measures.

| Embed\Batch | 32 | 128 |
|---|---|---|
| 25 | **5.08 ± 0.322** | 5.50 ± 0.414 |
| 100 | 5.95 ± 0.498 | 6.80 ± 1.15 |

Table 2: MSE across embedding dimensions (rows) and batch sizes (columns).

## B APPENDIX: DATASETS AND FEATURE EXTRACTORS

Exact model versions and pretrained weights are specified in the included GitHub repository. We ensured that at least two distinct feature extractors were chosen per unstructured modality (text, audio, and image) to demonstrate the generalizability of our proposed algorithm.

**RSNA Bone Age Prediction** The Radiological Society of North America (RSNA) Pediatric Bone Age Machine Learning Challenge collected pediatric hand radiographs labeled with the age of the subject in months Halabi et al. (2019). We collected 14,036 images from this dataset. Images were resized to 224x224 pixels, normalized with mean and standard deviation of 0.5 across the single gray-scale channel and input to ResNet-18 pretrained on ImageNet He et al. (2016); He and Jiang (2021). [1]

**ADReSSo** The Alzheimer's Dementia Recognition through Spontaneous Speech only (ADReSSo) diagnosis dataset has 237 audio recordings of participants undergoing the Cookie Thief cognitive

---

[1] https://pytorch.org/hub/pytorch_vision_resnet/

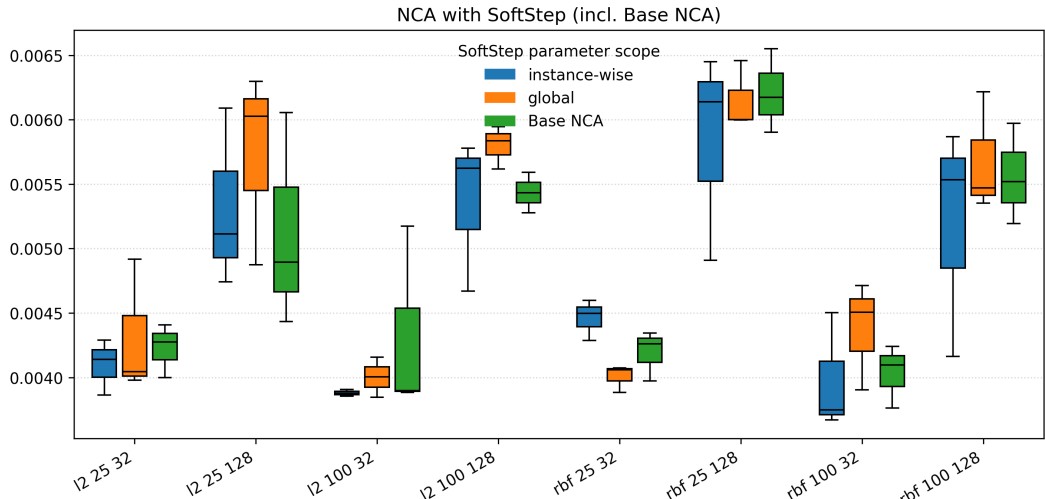

Figure 6: The x-axis ticks are hyperparameter combination triplets of the form (similarity measure, embedding dimension, batch size). The best-performing hyperparameter configurations by mean performance were: Base NCA (RBF, 100, 32), NCA with global SoftStep parameters ($\ell_2$, 100, 32), and NCA with instance-wise SoftStep parameters (RBF, 100, 32).

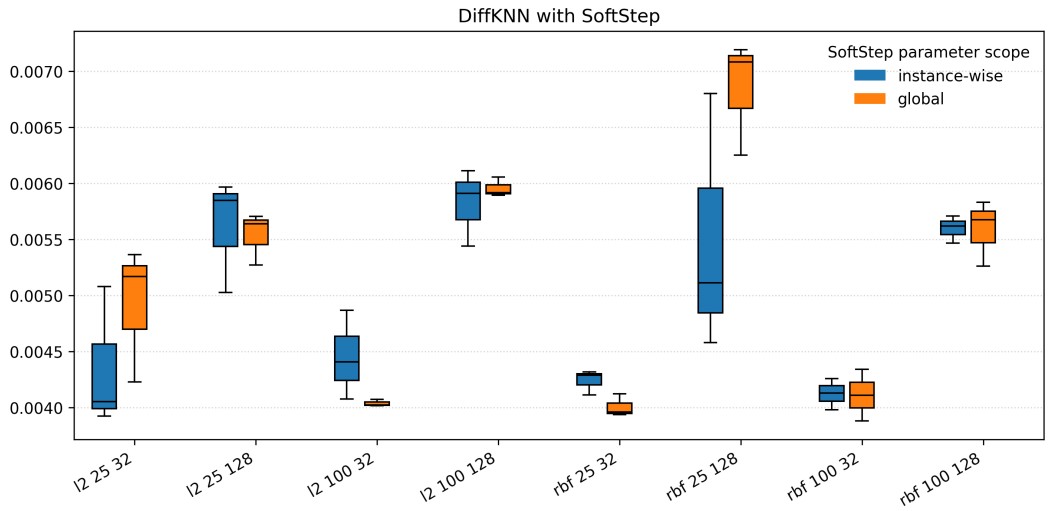

Figure 7: The x-axis ticks are hyperparameter combination triplets of the form (similarity measure, embedding dimension, batch size). The best-performing hyperparameter configurations by mean performance were: DiffKNN with global SoftStep parameters (RBF, 25, 32), DiffKNN with instance-wise SoftStep parameters ($\ell_2$, 25, 32).

assessment labeled with their score on the Mini Mental State Exam Luz et al. (2021b). Transcripts of these recordings were tokenized and input to DistilBERT-base-uncased Sanh (2019); Zolnoori et al. (2023).[2]

**MedSegBench**   The MedSegBench BriFiSegMSBench dataset is comprised of 1,360 single-channel microscopy images and corresponding segmentation masks Kuş and Aydin (2024). We estimated the size of segmentation mask areas using EfficientNet trained on ImageNet. Rizk et al. (2014); Tan and Le (2019).[3]

---

[2]https://huggingface.co/docs/transformers/en/model_doc/distilbert
[3]https://docs.pytorch.org/vision/main/models/efficientnet.html

**CoughVid**    The CoughVid dataset provides over 25,000 crowdsourced cough recordings, with 6,250 recordings labeled with participant age in years Orlandic et al. (2021). One second of cough audio was input to HuBERT pretrained on LibriSpeech and mean-pooled across time Hsu et al. (2021); Feng et al. (2024).[4]

**NoseMic**    The NoseMic dataset collected 1,297 30-second audio recordings of heart rate-induced sounds in the ear canal using an in-ear microphone under several activities Butkow et al. (2024). Audio clips were denoised,[5] encoded with the Whisper tiny audio encoder (Radford et al., 2023), and mean-pooled across time.[6]

**Udacity**    The Udacity self-driving car dataset is comprised of dashcam videos labeled with the angle of the car's steering wheel Du et al. (2019). Videos were downsampled to 4 frames per second for a total of 6,762 images and individual frames were input to EfficientNet trained on ImageNet Tan and Le (2019).[3]

**Pitchfork**    24,649 reviews from the website Pitchfork were collected (Pinter et al., 2020), where albums are scored from 0 to 10 in 0.1 increments. 1,500 randomly selected reviews were tokenized and input to BERT base Warner et al. (2024).[7]

**Houses**    The Houses dataset collects 535 curbside images of houses as well as their log-scaled list price Ahmed and Moustafa (2016). Images were resized to 256x256 pixels, center cropped to 224x224, ImageNet normalized, and input to ResNet-34 He et al. (2016).[1]

**Books**    The MachineHack Book Price Prediction dataset collated 6237 synopses of books labeled with their log-normalized price.[8] Synopses were tokenized and input to DistilBERT-base-uncased.[2]

**Austin**    The Kaggle Austin Housing Prices dataset collects over 15000 descriptions of homes labeled with their log-scaled list price.[9] Descriptions were tokenized and input to DistilBERT-base-uncased.[2]

**Wiki-IMDB**    The Wiki-IMDB dataset collects 523,051 dated photographs of celebrity faces as well as their birthdays allowing for age prediction Rothe et al. (2018). 101,590 of these images were resized to 224x224 pixels, ImageNet normalized, and input to ConvNeXt base. Liu et al. (2022).[10]

## C    APPENDIX: HIGHER ORDER EMBEDDING STRUCTURE WITH NEIGHBOR-BASED PREDICTION + MSE

We showed through an argument in Section 4 that the combination of neighbor-based prediction and MSE loss implicitly imposes optimization conditions on all pairs and triplets of embeddings. However, the setup of our argument is general for substructures of all orders within the embedding space. It is natural then to explore whether our argument yields embedding conditions on such higher order structures. We briefly show here that the analogous argument does not yield a unique embedding condition for structures beyond triplets regardless of label order.

For some $i \in \{1, ..., N\}$, and an indexing set $J = \{j_1, j_2, ..., j_M\} \subset \{1, ..., N\}$, we seek to minimize $T_{iJ} = (\sum_{m=1}^{M} \Delta_{ij_m} p_{ij_m})^2$. As in our earlier argument we assume no two labels in the neighbor set being considered are equal, and, without loss of generality, if $m_1 < m_2$, then $y_{j_{m_1}} < y_{j_{m_2}}$. Additionally, no labels in the neighbor set are equal to $y_i$. If $R = \sum_{m=1}^{M} p_{ij_m}$ as before, then this is a convex quadratic optimization with respect to the vector $p_i^* = (p_{ij_1}, ..., p_{ij_M})$ over the standard simplex scaled by $R$. Let $\{e_1, ..., e_M\}$ be the standard basis vectors in $\mathbb{R}^M$ and define $\lambda = \frac{\Delta_{ij_{m+1}}}{\Delta_{ij_{m+1}} - \Delta_{ij_m}}$. Immediately, we see that there is a solution to our optimization given by the vector

$$p_i^* = \begin{cases} R \cdot e_1, & y_i < y_{j_1} \\ R \cdot e_M, & y_i > y_{j_M} \\ \lambda R \cdot e_m + (1 - \lambda) R \cdot e_{m+1}, & y_{j_m} < y_i < y_{j_{m+1}}, \text{ for some } 1 < m < M \end{cases}$$

---

[4]https://huggingface.co/facebook/hubert-base-ls960
[5]https://pypi.org/project/noisereduce
[6]https://huggingface.co/openai/whisper-tiny
[7]https://huggingface.co/google-bert/bert-base-uncased
[8]https://machinehack.com/hackathons/predict_the_price_of_books/overview
[9]https://www.kaggle.com/datasets/ericpierce/austinhousingprices
[10]https://docs.pytorch.org/vision/main/models/convnext.html

The case of interest, where a nontrivial optimal structuring condition may be found, is the final one. However, this case yields the same triplet condition as in the original argument. This proves that an additional condition for higher order structures is not guaranteed by a partial optimization of $T_{i,J}$. While this argument does not preclude the possibility of higher order structuring, further insight is needed to uncover such conditions.

## D  APPENDIX: NEIGHBOR-BASED CLASSIFICATION

In section 4, we demonstrate that pairing neighbor-based regression with a mean square error objective empowers models to learn well-structured, semantically rich, and predictive embedding spaces. Additionally, the inclusion of learned sparse and adaptive similarity measures (e.g., SoftStep) in such models further promotes the realization of these embedding spaces. We note here that in the context of classification with cross entropy-style losses, a parallel analysis to the one we conduct in section 4 does not yield the same conclusions.

Given an embedded sample of data, $z$, the probability associated with $y(z)$ is

$$p(y(z)) = \frac{1}{\sum_{j=1}^{N} \exp(s_j)} \sum_{\substack{i=1 \\ y_i = y(z)}}^{N} \exp(s_i)$$

where each $s_i$ is some similarity measure between the $i$-th sample of the neighbor set (of size $N$) and $z$. Therefore, the cross entropy objective, $-\log(p(y(z)))$, reduces to maximizing the proportional similarity associated with samples from the same class. It is known within the metric learning literature that this style of objective suffers from the problem of class collapse Graf et al. (2021); Papyan et al. (2020). In an unbounded space where there is no minimum proportional similarity associated with sampled in different classes, there is the additional complication of the model learning to embed samples from different classes as dissimilarly as possible, which can promote gradient explosions and instability in training. Specifically,

$$\lim_{\min\{s_i : y_i \neq y_z\} \to 0} p(y(z)) = 1$$

Furthermore, inference within this framework is highly subject to class imbalance. It is possible that the inclusion of sparse similarity (e.g. SoftStep) may solve some of these issues, as the objective is relaxed to only embed samples from the same class in the learned neighborhood of $z$. Connections to metric learning also suggest that the viability of this approach can be restored with more sophisticated metric learning-style objectives. However, this presents a large research challenge in its own right and is beyond the scope of this work.

