# OpenReview forum: "SoftStep: Learning Sparse Similarity Powers Deep Neighbor-Based Regression"
_ICLR.cc/2026/Conference — ICLR 2026 Conference Desk Rejected Submission_

### Official Review · Reviewer_N2ZZ · 2025-10-28

**Soundness:** 2
**Presentation:** 2
**Contribution:** 2
**Rating:** 4
**Confidence:** 4

**Summary:**

The paper introduces an module named SoftStep, designed to learn sparse, sample-adaptive similarity measures in neural networks and integrates it with a nearest-neighbor-based regression approach. The core idea is novel, supported by in-depth theoretical analysis, and demonstrates potential advantages over linear regression heads across multiple datasets. However, the paper exhibits notable limitations in the breadth and depth of experiments, evaluation of computational efficiency, and the completeness of certain theoretical analyses, which currently affect its maturity and acceptability.

**Strengths:**

1) SoftStep provides a learnable sparse similarity metric, which differs from existing methods that rely on fixed sparsity patterns.

2) The paper derives implicit geometric constraints from the MSE loss, revealing the structural properties underlying neighbor-based regression.

3) Experimental results show some improvements over linear heads and traditional NCA/kNN methods across multiple tasks.

**Weaknesses:**

1) The experimental evaluation appears limited in scope, as it lacks comparisons with contemporary sparse attention mechanisms such as Sparsemax and Sparse Transformer. The current baseline methods—limited to linear attention and NCA—are insufficient to comprehensively validate the method's advantages. Furthermore, the proposed approach includes multiple variants that complicate the interpretation of results and dilute the clarity of the key contributions.

2) The method is only validated on regression tasks, leaving open questions about the method's efficacy in classification and unsupervised learning settings. A broader discussion is suggested to establish general applicability.

3) The effects of SoftStep parameters (l, u, t) and sparsity variation have not been analyzed. Without proper ablation studies and sensitivity analysis, the robustness of the proposed method remains unverified. The absence of discussion on parameter selection guidelines significantly limits the practical utility of this work.

4) The paper omits a critical analysis of computational efficiency, including training/inference speed, memory footprint, and scalability. Without these metrics, the practical utility of SoftStep for real-world applications remains uncertain, particularly for resource-constrained environments or large-scale deployments.

**Questions:**

1) Can SoftStep be applied to classification tasks? What modifications would be required to adapt it for that setting?

2) Compared with existing sparsification methods such as Sparsemax, what are the main advantages of SoftStep?

3) How are the parameters $l$, $u$, $t$ of SoftStep initialized and updated during training? Are there any issues of numerical instability or convergence?

4) In Table 1, the NCA-i method shows large performance variance on the MedSegBench dataset. What could be the possible reasons for this instability?

5) What is the relationship between the SOFTKSTEP function in the algorithm and the SoftStep function defined in Equation (1)? Why do they use different parameterizations?

---

> ### Author Response · Authors · 2025-11-21
>
> Dear reviewer N2ZZ,
>
> Thank you for the time and effort you put into reviewing our paper, and for the very insightful comments. Here are our responses to your observations and questions.
>
> 1) In the final revision’s supplementary materials, we will add an analysis showing that neighbor-based methods with cross-entropy loss effectively reduce to learning linear separability, and therefore do not offer the same representational benefits as in the MSE-based regression setting. That said, the connection to metric learning objectives in regression suggests that similar gains may be attainable in other task settings with appropriate losses, and we will adjust the discussion language around classification to reflect this.
>
> 2) We agree completely about expanding our evaluation and have included sparsemax as a new benchmark on all datasets. Our interpretation of the results is that, almost universally, the inclusion of SoftStep enables neighbor-based prediction over benchmark approaches and the standard linear head. Variation between NCA with SoftStep vs DiffKNN are more marginal and attributable to differences between datasets. We have rewritten the language of the results section to make this more clear.
>
> 3) In our experiments we compare global SoftStep parameters to instance-wise learned parameters. In each of these settings, the parameters are initialized with respect to standard initializations (unit normal initialization for global and standard linear layer initialization for instance-wise). We did not encounter instabilities during optimization, as small epsilons are included in the code to avoid division by zero or rejection of all neighbors.
>
> 4) Efficiency and scaling:
>
>     a) We agree completely about the need for a computational complexity/efficiency analysis. We have added empirical results comparing the computational overhead of each method and the final revised version of the paper will include a complexity analysis that will describe the asymptotic costs of our methods with respect to model dimension and size of the neighbor set used for predictions.
>
>     b) In the current revision, we’ve included results on the Wiki-IMDB dataset, a dataset of 100k+ samples of faces that we’ve used for age prediction. Your observation about the difficulty of scaling is absolutely correct, but can be addressed by selecting a representative subset of the training set to use as neighbors as inference. We have included language describing this procedure in the revision.
>
> 5) Your observation about the results on the MedSegBench dataset is very astute! When inspecting the fold-level performance in the 10 fold cross-validation, we found a large spread in MSE - some folds were competitive with the rest of the evaluated algorithms, but most were far higher. This led to large variance. This does seem to be an interesting outlier in terms of instance-wise learning. It is possible that the increased parameter count of the instance-wise formulation could leave a model more prone to overfitting compared to learning one global parameter set. However, this trend is not borne out in the other 10 evaluations. Future investigations into what drove this behavior could certainly improve robustness, thank you!
>
> 6) The SoftKStep function is just a flipped version of the standard SoftStep function since it acts on similarity rankings where lower values correspond to higher rankings. In the first revision, we have included language to clarify this.

---

### Official Review · Reviewer_AKZd · 2025-11-01

**Soundness:** 3
**Presentation:** 3
**Contribution:** 2
**Rating:** 2
**Confidence:** 4

**Summary:**

- SoftStep proposes a differentiable, neighbor-based regression head that learns sparse similarities with learnable \ell,u,t. it can be easily plugged into soft kNN / NCA for end-to-end training.
- The pipeline: pretrained encoder → small MLP to an embedding → SoftStep → neighbor-weighted prediction.
- The claim is that replacing the usual linear head with SoftStep-based neighbor regression improves MSE and yields more structured embeddings.
- Experiments span several regression datasets (vision / audio / text), reporting consistent gains over linear heads and vanilla NCA/kNN.
- However, despite stating easy applicability to diverse architectures, the paper largely evaluates on limited backbones and scales, leaving generality and efficiency underexplored.

**Strengths:**

- Clear, modular formulation: SoftStep is a drop-in, differentiable sparsifier for neighbor heads.
- Empirical results are consistently better than linear regression heads (and vanilla neighbor baselines) across multiple regression datasets.
- Theoretical intuition is reasonable: neighbor-based MSE induces pair/triplet structure.
- The method exposes meaningful knobs ((\ell,u,t); global vs instance-wise) that could be useful for controlling sparsity and locality.

**Weaknesses:**

- Backbone/scale generalization is thin: claims of easy applicability are not substantiated across diverse and larger encoders (e.g., ViT/ConvNeXt/BERT) or large-scale datasets; results remain small to mid-scale.
- Comparisons are not strong enough: lacks head-to-head against robust modern alternatives for regression/metric learning (e.g., recent metric learning algorithms).
- Attribution is unclear: separate SoftStep from soft-rank effects: run soft-rank on/off ablations, and test rank-free NCA with SoftStep only to see whether gains persist without soft-rank.
- Efficiency and scaling are underreported: neighbor heads can be costly.

**Questions:**

- Can you evaluate on diverse backbones and larger benchmarks (e.g., ViT/ConvNeXt for vision, BERT/RoBERTa for text; ImageNet-scale or comparable regression tasks) to substantiate the “easy to apply” claim?
- Under identical training, how does SoftStep compare with strong regression/metric baselines?
- Please provide isolation ablations: (i) SoftStep on/off with identical soft-kNN/NCA, (ii) global vs instance-wise (\ell,u,t) to identify what actually drives the gain.
- (If applicable) Could you include non-regression tasks (classification/ranking) to demonstrate broader utility beyond the current regression focus?

---

> ### Author Response · Authors · 2025-11-21
>
> Dear reviewer AKZd,
>
> Thank you for the time and effort you put into reviewing our paper, and for the very insightful comments. Here are our responses to your observations and questions.
>
> 1) In the current revision, we’ve included results on the Wiki-IMDB dataset, a dataset of 100k+ samples of faces that are used for age prediction. Your observation about the difficulty of scaling is absolutely correct, but can be addressed by selecting a representative subset of the training set to use as neighbors as inference. We have included language describing this procedure in the revision.
>     - Our experimentation includes large models such as BERT and Whisper (Appendix section B) and we used ConvNeXt for the Wiki-IMDB set to further diversify our library of backbone models.
>
> 2) Thank you for your point about metric learning. This is essential for us to clarify. We feel it necessary to include a background section on metric learning as our work is indebted to traditional metric learning, e.g. NCA and LMNN. We also feel it necessary to include references to sparse similarity measures in metric learning.  However, we did not sufficiently emphasize that we don't compare to other metric learning algorithms since our primary focus is a novel regression head for neural networks. We simply meant to demonstrate that our regression head enforces a metric implicitly, without additional algorithmic burden. We apologize for the confusion and have updated the language of our submission accordingly.
>
> 3) Soft-rank is not separable from SoftStep in this formulation, as we use SoftStep as a proxy for estimating the “k” parameter in k-NN. The alternative to SoftStep would be to pre-select a fixed, integer “k” parameter and apply it after computing rankings. However, optimal tuning of this integer “k” would require an unreasonable computational burden as it requires a grid search over the batch dimension (where each iteration fully trains a neural network). Instead, SoftStep enables learning “k” during training without a grid search. We include the other major ablation you mention - NCA without SoftStep - as NCA is a stand alone prediction method.
>
> 4) We agree completely with your point regarding reporting efficiency. We have added empirical results comparing the computational overhead of each method and the final revised version of the paper will include a complexity analysis that will describe the asymptotic costs of our methods with respect to model dimension and size of the neighbor set used for predictions.
>
> 5) In the final revision’s supplementary materials, we will add an analysis showing that neighbor-based methods with cross-entropy loss effectively reduce to learning linear separability, and therefore do not offer the same representational benefits as in the MSE-based regression setting. That said, the connection to metric learning objectives in regression suggests that similar gains may be attainable in other task settings with appropriate losses, and we will adjust the discussion language around classification to reflect this.

---

### Official Review · Reviewer_KjUS · 2025-11-01

**Soundness:** 2
**Presentation:** 2
**Contribution:** 3
**Rating:** 4
**Confidence:** 4

**Summary:**

This paper proposes a differentiable regression head which, when paired with soft kNN or neighborhood component analysis operations, allows learning a nearest-neighbor based classifier at the end of a neural network. The authors show that, on a variety of datasets, this neighborhood-based classification outperforms linear regression heads.

**Strengths:**

The idea itself is quite neat. As I understand it, the authors allow for learning a smooth function over the number of neighbors to use when doing predictions. It is also nice that it works with both the differentiable knn and the neighborhood component analysis set ups. The presentation in the first three sections in particular is very easy to follow.

**Weaknesses:**

I think the paper has three primary weaknesses.

The first is that section 4 is quite strange in how it is presented. It seems that the authors are trying to make theoretical statements verifying that their approach works. However, it's not clear what precisely what is being shown and it seems there are some mistakes in this section? For example, the phrase "we demonstrate that a neighbor-based regression model paired with MSE loss yields implicit optimization conditions for structuring pairs of points in the embedding space with respect to their labels" is never formalized. It would be good if there was a theorem or lemma statement which specified precisely what the authors are proving, followed by a clear proof. The same goes for the bolded sentences in section 4, these should be stated as corollaries and proven. Regarding the mistakes, I'm confused why the authors start discussing terms of the form $\Delta_{ij}^2 p_{ij}^2$ on line 308. The previous equation had $\sum_i \left( \sum_j \Delta_{ij}^2 p_{ij} \right) ^2$. These are different things. It's also not clear to me where the equation for the optimal triplet embedding geometry is coming from. It's also not clear what we are trying to do with it or how it relates to the softstep method. Again, these statements would be easier to interpret if there were formal theorem/lemma/proposition/corollary statements which made them clear.

The second weakness is that the experiments are confusing and the results are hard to interpret. Below is a non-exhaustive list of questions which I have after looking through the results:
- Why are these the datasets which were used? Why not MNIST, CIFAR10 and Imagenet for computer vision tasks and/or standard NLP ones for NLP tasks?
- What are the backbone architectures being optimized?
- Why was trainin always done using pre-trained networks rather than training ones from scratch with the softstep algorithm? What happens if you train from scratch?
- How can NCA g ever outperform NCA i? Isn't NCA g simply a subset of the expressivity of NCA i? Same for DiffKNN.

Finally, the third weakness of the paper is that the utility of softstep is not made fully clear. As I understand it, when paired with a radial regression algorithm, softstep allows learning the parameters which control the radii. If this is the case, then it seems like it should be compared against more than just a linear classifier. Specifically, we should compare against other known regression algorithms that can do similar radial, nonlinear classification and regression tasks. For example, what happens if we simply use the softrank and  NCA algorithms without softstep? Or differentiable PSD kernels? Similarly, why do we require the logarithm when applying softstep into $Sim(Z, Z_N)$?

In summary, the above weaknesses make it difficult to judge the paper's presented algorithm fully. The theoretical claims are not sufficiently clear and the experimental analysis leaves many questions unanswered. Although the work is developing an interesting idea, I do not think the analysis is sufficiently robust to convince a reader that the method is clearly superior.

**Questions:**

See the above discussion.

---

> ### Author Response · Authors · 2025-11-21
>
> Dear reviewer KjUS,
>
> Thank you for the time and effort you put into reviewing our paper, and for the very insightful comments. Here are our responses to your observations and questions.
>
> 1) Thank you for pointing out improvements that we can make in Section 4. The main idea is to show the underlying embedding geometry that drives the performance of neighbor-based regression and how the inclusion of sparse similarity measures with SoftStep enables the realization of this optimally predictive geometry. The triplet conditions are also made achievable by the adaptive/asymmetric similarity of our method. We are working on formalizing the presentation of math for the final revision to demonstrate this more clearly.
>      - The derivation on line 308 comes from expanding the expression $(\sum_{j=1}^N \Delta_{ij}p_{ij})^2$ and focusing on the partial optimization of the like terms.
>
> 2) We didn’t use more standard datasets because the proposed method is for regression in deep learning. Most standard deep learning datasets have categorical labels, especially in computer vision. In the final revision’s supplementary materials, we will add an analysis showing that neighbor-based methods with cross-entropy loss effectively reduce to learning linear separability, and therefore do not offer the same representational benefits as in the MSE-based regression setting. That said, the connection to metric learning objectives in regression suggests that similar gains may be attainable in other task settings with appropriate losses, and we will adjust the discussion language around classification to reflect this.
>
> 3) All of the backbones in our experiments are detailed in section B of the appendix. They include CNNs (now including ConvNeXt!), text transformers, audio transformers.
>
> 4) We agree completely on your point about training from scratch. We are currently running experiments to include results on networks trained from scratch to address this. We leaned towards fine-tuning originally simply because it reflects common real world workflows for solving prediction problems with deep learning. However, we agree that it is important to demonstrate that our method works in the randomly initialized setting as well.
>
> 5) Your observation about the global formulation outperforming the instance-wise formulation is very astute. The most likely explanation for this is model overfitting. In certain contexts without careful regularization, the increased parameterization of the instance-wise formulation may overfit to the training set.
>
> 6) SoftStep is not paired with radial regression in this work. Radial regression learns Gaussian parameters to transform representations before applying linear regression, whereas our approach predicts directly from embedding similarities (optionally using an RBF kernel). SoftStep warps these similarities to make them more predictive, as discussed in Section 4. Most alternative regression methods, like those you mention, operate by learning a nonlinear representation and then applying linear regression. This type of formulation does make them compatible with our framework, as we can simply replace their linear head with our neighbor-based one. For this reason, we compare primarily against linear regression under matched backbones. We also acknowledge other ways to induce sparse similarities, such as sparsemax, and have added sparsemax as a benchmark in the revised paper.
>     - We add the log of the SoftStep outputs as a mathematical trick so that when softmax is applied, the composition of the operations scales the similarities by the original SoftStep outputs.

---

### Official Review · Reviewer_PMLH · 2025-11-02

**Soundness:** 4
**Presentation:** 3
**Contribution:** 3
**Rating:** 6
**Confidence:** 3

**Summary:**

This paper proposes replacing the linear output head with an output head based on differentiable nearest neighbors.

In order to do this, the paper proposes the SoftStep architecture, which is combined with NCA or differentiable k-nearest neighbors.

This allows the network to jointly learn an embedding, a sparse set of nearest neighbors, and output labels corresponding to a weighted average of the labels on these neighbors.

**Strengths:**

This paper proposes a new read-out head architecture that appears to have better performance than linear read-out on a suite of benchmark datasets.

The method is novel to the best of my knowledge.

The method is well motivated, and generally well presented.

**Weaknesses:**

* A nearest neighbor head seems much harder to scale in terms of number of data points than a linear head. New ideas seem to be needed to make this method work for larger datasets. Accordingly, the experiments are on relatively smaller scale benchmarks.

* The presentation was generally good, but I was confused with some aspects: see my questions below.

**Questions:**

* I didn't understand what is meant by global vs. model-level parameters for SoftStep. Could you please

* What is the asymptotic cost of this method in terms of number of data points? Is it quadratic in the dataset size? That could be good to include. More generally, is this method limited to smaller datasets for now?

---

> ### Author Response · Authors · 2025-11-21
>
> Dear reviewer PMLH,
>
> Thank you for the time and effort you put into reviewing our paper, and for the very insightful comments. Here are our responses to your observations and questions.
>
> 1) In our paper, “model-level” and “global” parameters refer to the same formulation. We apologize for the imprecise language and have clarified our language in our first revision.
>
> 2) We agree completely about the need for a computational complexity/efficiency analysis. We have added empirical results comparing the computational overhead of each method, and the final revised version of the paper will include a complexity analysis that will describe the asymptotic costs of our methods with respect to model dimension and size of the neighbor set used for predictions.
>
> 3) In the current revision, we’ve included results on the Wiki-IMDB dataset, a dataset of 100k+ samples of faces that are used for age prediction. Your observation about the difficulty of scaling is absolutely correct, but can be addressed by selecting a representative subset of the training set to use as neighbors as inference. We have included language describing this procedure in the revision.

---

### Author Response · Authors · 2025-12-03
**Area Chair response 1**

Dear Area Chair,

We truly thank you for reading our rebuttals, and appreciate your extra effort given these unfortunate circumstances.

Our reviewers provided very helpful critiques, which we have addressed completely and which have made our work much stronger. We believe the bulk of these critiques were necessary but custodial, and did not take issue with the theory or implementation of our methods, nor the strength of contribution. As such we believe our revised manuscript is now easier to read and our methods are more holistically evaluated, befitting acceptance to ICLR.

We summarize and categorize the critiques as follows: Computational Analysis, Expansion of Evaluation, Comparison to Baselines, Extension to Classification, and Misunderstandings (response 2).

Computational Analysis
Reviewers PMLH, AKZd, and N2ZZ asked for analytical and empirical computational complexity analysis. This includes big O analysis as well as FLOP counts per algorithm tested
Reviewer PMLH: “What is the asymptotic cost of this method in terms of number of data points? Is it quadratic in the dataset size?”
Reviewer AKZd: “Efficiency and scaling are underreported: neighbor heads can be costly.”
Reviewer N2ZZ: “The paper omits a critical analysis of computational efficiency, including training/inference speed, memory footprint, and scalability.”

Figure 4 provides the observed FLOP counts of a forward and backwards pass with each method under identical conditions and addresses the asymptotic cost

Expansion of Evaluation
Reviewers KjUS and AKZd asked us to expand our evaluation to larger datasets as well as train models from scratch (as opposed to supervised fine-tuning pre-trained models)
Reviewer KjUS: “Why was trainin always done using pre-trained networks rather than training ones from scratch with the softstep algorithm? What happens if you train from scratch?”
Reviewer AKZd: “Backbone/scale generalization is thin: claims of easy applicability are not substantiated across…large-scale datasets; results remain small to mid-scale.”

Table 1 now includes new evaluations that were requested
We include the Wiki-IMDB dataset, which contains 101,590 images each labeled with the age of the photographed person
We train randomly initialized CNN models from scratch on the RSNA and Wiki-IMDB datasets
We did not train randomly initialized transformer-based models from scratch as they require self-supervised pretraining, which our datasets cannot support without overfitting

Comparison to Baselines
Reviewer N2ZZ pointed out that Sparsemax is an integral sparse attention mechanism that warrants comparison to this work
Reviewer N2ZZ: “The experimental evaluation appears limited in scope, as it lacks comparisons with contemporary sparse attention mechanisms such as Sparsemax”

We included Sparsemax in our evaluation by incorporating it with NCA, ran hyperparameter tuning, and applied it to the full evaluation
Sparsemax won in 1 of 13 evaluations, confirming the usefulness of this critique as well as the superiority of our proposed method

Extension to classification:
Reviewers AKZd, N2ZZ asked us to expand our methodology to classification.
Reviewer AKZd: “Could you include non-regression tasks (classification/ranking) to demonstrate broader utility beyond the current regression focus?”
Reviewer N2ZZ: “Can SoftStep be applied to classification tasks?”

Differentiable neighbor-based methods for classification (NCA) fail due to class collapse and exposure to batch effects, as we demonstrate in Appendix D. It is possible that the inclusion of sparse similarity (e.g. SoftStep) may solve some of these issues. However, we maintain this presents a large research challenge in its own right and is beyond the scope of this paper.

---

### Author Response · Authors · 2025-12-03
**Area Chair response 2**

Continued...

Misunderstandings
Reviewer AKZd suggested we use BERT and Transformer models, which we did use and documented in our paper
Reviewer AKZd: “Can you evaluate on diverse backbones and larger benchmarks (e.g., ViT/ConvNeXt for vision, BERT/RoBERTa for text”
The appendix states the backbones we used, which heavily include BERT and ConvNeXt

Reviewer KjUS asks which backbones we use, which is clearly documented in our paper
Reviewer KjUS: “What are the backbone architectures being optimized?”
The appendix states the backbones we used and the specifics of every pipeline

Reviewer AKZd suggested an ablation using NCA with and without SoftStep/global vs instance-wise, which we did perform and documented in our paper
Reviewer AKZd: “Please provide isolation ablations: (i) SoftStep on/off with identical soft-kNN/NCA, (ii) global vs instance-wise (\ell,u,t) to identify what actually drives the gain.”
The main results table (Table 1) clearly shows performance comparisons of all these ablations.

Reviewer KjUS misread our mathematical analysis
Reviewer KjUS: “...on line 308. The previous equation had $\sum_i \big(\sum_j\Delta_{ij}^2p_{ij} \big)^2$. These are different things.”
The reviewer adds an extra exponent in their quotation that is the source of their confusion. The actual term is $\sum_i \big(\sum_j\boldsymbol{\Delta_{ij}}p_{ij} \big)^2$ (line 301)

Reviewer KjUS calls our method “radial regression” and suggests we compare to appropriate regression algorithms (unfortunately without citations); we do not use radial regression in this work
Reviewer KjUS: “As I understand it, when paired with a radial regression algorithm, softstep allows learning the parameters which control the radii”
Radial regression reduces to a learned z-scoring, where features are projected onto a learned multivariate Gaussian, and then input to a linear regression head.
This is not what we propose. We propose a neighbor-based regression head, which is demonstrated to consistently outperform a linear regression head in head-to-head comparison
We note here that promising regression techniques in the literature make a final regression estimate using a linear regression head, despite interesting upstream feature projections.
Walczak, B., and D. L. Massart. "The radial basis functions—partial least squares approach as a flexible non-linear regression technique." Analytica Chimica Acta 331.3 (1996): 177-185.
Jiang, Xiangjian, et al. "Protogate: Prototype-based neural networks with global-to-local feature selection for tabular biomedical data." arXiv preprint arXiv:2306.12330 (2023).
Yang, Junchen, Ofir Lindenbaum, and Yuval Kluger. "Locally sparse neural networks for tabular biomedical data." International Conference on Machine Learning. PMLR, 2022.
Hesamian, Gholamreza, Arne Johannssen, and Nataliya Chukhrova. "Fuzzy nonlinear regression modeling with radial basis function networks." IEEE transactions on fuzzy systems 32.4 (2023): 1733-1742.

Reviewers KjUS asks for inclusion of classification-task datasets like MNIST, despite our work focusing on regression.
Reviewer KjUS: “Why are these the datasets which were used? Why not MNIST, CIFAR10 and Imagenet for computer vision tasks and/or standard NLP ones for NLP tasks?”
We propose “SoftStep: Learning Sparse Similarity Powers Deep Neighbor-Based Regression”

Reviewer AKZd says we lack comparisons to metric learning approaches, which is a misreading of our work
Reviewer AKZd: “Comparisons are not strong enough: lacks head-to-head against robust modern alternatives for regression/metric learning (e.g., recent metric learning algorithms).”
We have revised the language of our submission to clearly note that we do not propose a metric learning algorithm, but rather induce metric learning-like structuring “for free” by pairing neighbor-based regression with mean-squared error loss
Most metric learning approaches apply auxiliary loss functions to structure a latent space, and then use a linear head for final regression.

---

### Note · Program_Chairs · 2026-01-17
**Submission Desk Rejected by Program Chairs**

The following references in this submission do not refer to real documents and/or have major errors in bibliographic information:

 Zhiyi Su, Yulia R. Gel, and Joseph L. Gastwirth. The chngpt r package for threshold regression modelling. Journal of Statistical Software, 76(1):1-32, 2017.
Hussein Hazimeh and Rahul Mazumder. Fast and accurate sparse regression via oracle. Proceedings of the 37th International Conference on Machine Learning, 119:4207-4216, 2020.
R. J. Batista, C. A. Silva, and I. G. Costa. Efficient epistasis detection via cartesian interaction modelling. Bioinformatics, 40(5):831-838, 2024.